# Factors for arboviral seropositivity in children in Teso South Sub County, Kenya

Mary Inziani[1]*, Jane Mawia Kilonzo[2], Marthaclaire Kerubo[2], Sylvia Mango[3], Mary Kavurani[2], Allan Ndirangu[2], Elizabeth Njeri[2], Diuniceous Oigara Ogenche[3], Sylvester Ogolla Ayoro[3], Shingo Inoue[4,5], Kouichi Morita[5], Matilu Mwau[2,3]

1 Centre for Virus Research, Kenya Medical Research Institute, Nairobi, Kenya, 2 KEMRI Laboratory for Molecular Biology, Kenya Medical Research Institute, Nairobi, Kenya, 3 Centre for Infectious and Parasitic Diseases Control Research, Kenya Medical Research Institute, Alupe, Busia, Kenya, 4 Nagasaki University Africa Research Station, Nairobi, Kenya, 5 Nagasaki University Institute of Tropical Medicine, Nagasaki, Japan

* mmuyeku@kemri.go.ke, inzianimatilu@gmail.com

## Abstract

### Background

Arboviruses like Yellow Fever Virus (YFV), Dengue Virus (DENV), Chikungunya Virus (CHIKV), and West Nile Virus (WNV) frequently cause outbreaks in sub-Saharan Africa. Identifying risk factors in children can improve diagnosis, treatment, and prevention strategies. This study identified factors associated with seropositivity to YFV, DENV, CHIKV and WNV among children in Teso South Sub-County, Western Kenya.

### Methods

This survey involved 656 children aged 1–12 years, enrolled at two health facilities. Socio-demographic, environmental, behavioral, and medical information was collected via a questionnaire. Serological screening for antibodies to YFV, DENV, CHIKV, and WNV was performed using Indirect Enzyme-Linked Immunosorbent Assays. The collected data was summarized using descriptive statistics. Factors associated with seroprevalence were examined using multinomial logistic regression.

### Results

Overall, 27.7% of children were seropositive for at least one arbovirus: 15.7% for DENV, 9.6% for WNV, 5.6% for CHIKV, and 4.4% for YFV. Factors associated with any arbovirus were: female gender, age 6–9 and 9–12 years, non-parent primary caregiver, and use of unknown bed nets brand (p < 0.05). YFV seropositivity was not associated with any of the risk factors, while DENV was associated with female gender and age 6–9 years (p < 0.05). CHIKV was associated with use of insect repellents and not using any mosquito bed nets. WNV seropositivity was significantly higher in all children aged above 3 years, those who lived in town/urban areas, use of olyset,

**Data availability statement:** All relevant data are within the manuscript and its Supporting Information files.

**Funding:** The author(s) received no specific funding for this work.

**Competing interests:** The authors have declared that no competing interests exist.

supanet and unknown bed nets and in those who lived in houses roofed with tiles and iron sheets (p < 0.05).

## Conclusion

Arbovirus exposure among children is influenced by age, female gender, non-parental primary care giver, failure to use mosquito bed nets, type of bed net, use of insect repellents, and house roofing material. Interventions targeting housing improvements, education on bed net and mosquito repellent use, and environmental mosquito control can reduce infection risks in endemic areas.

## Introduction

Arthropod-borne viruses (arboviruses), such as Chikungunya (CHIKV), Yellow Fever (YFV), Dengue Virus (DENV), and West Nile Virus (WNV), are transmitted to humans primarily by mosquitoes. *Aedes species* are the primary vectors for CHIKV, YFV and DENV, while WNV is transmitted mainly by *Culex species* [1–9]. These viruses affect both adults and children, causing significant morbidity worldwide [10,11], and are responsible for frequent outbreaks, with many emerging and re-emerging in sub-Saharan Africa [12,13]. Kenya, in particular, has reported several arboviral outbreaks in recent years [4,6,7,14–16]. Children in endemic areas are especially vulnerable to these infections due to their developing immune systems, making them more susceptible to severe disease outcomes [17].

Arboviral infections often present with non-specific symptoms, including fever, jaundice, swollen lymph nodes [17], neuro-invasive disease [8,18], joint inflammation [19], and rashes [20], which can make clinical diagnosis challenging. In sub-Saharan Africa, arboviral infections frequently go un diagnosed and unreported [21], as febrile illnesses are commonly misdiagnosed as malaria, typhoid, or other bacterial infections [8,22]. In regions with limited diagnostic resources, malaria-negative fevers are often treated presumptively with antimalarials or antibiotics, further complicating accurate diagnosis and management [23].

Despite the high burden of arboviral diseases and their significant impact on public health in Kenya, there is limited research on the prevalence, epidemiology, and associated factors for these infections in children. A few studies have attempted to document the magnitude of arbovirus infections or identified key selected factors in this population [10,21,24–30]. This study aims to fill this gap by exploring the selected factors associated with seropositivity to YFV, DENV, CHIKV, and WNV among children in Teso South Sub County, Western Kenya. Understanding these selected factors is critical for improving early diagnosis, treatment, and the implementation of effective preventive measures in affected communities.

## Materials and methods

### Study design and setting

This hospital-based cross-sectional study was conducted between August 2010 and February 2011 in Teso Sub County, Western Kenya. We assessed the selected

factors for seropositivity to YFV, DENV, CHIKV, and WNV among children aged 1–12 years. A total of 656 children were recruited from those seeking health services at Alupe Sub County Hospital and KEMRI Alupe Clinic.

Data on socio-demographic, environmental, and clinical factors were collected using a structured questionnaire. Additionally, participants' clinical records and immunization cards were reviewed when available.

### Seropositivity testing

Approximately 2.5 mL of venous blood was collected from each participant for antibody testing. Virus-specific IgA/IgM/IgG Sero complex antibodies to YFV, DENV, CHIKV, and WNV were detected using an in-house Indirect Enzyme-Linked Immunosorbent Assay (ELISA) method, based on protocols described by Igarashi et al. [31,32] with minor modifications to suit local laboratory conditions [33]. The assay did not differentiate between the specific antibody sub-types (IgA, IgM, or IgG). The optical density (OD) of each sample was measured at 492 nm using an ELISA plate reader, within 20 minutes of adding the stop solution. Virus-specific OD was calculated as the difference between the mean OD of virus-coated wells and the mean OD of PBS-coated wells. A sample was considered seropositive if the OD was greater than 1.0. In cases where the blood volume was insufficient, priority for testing was given to YFV, followed by DENV2, CHIKV, WNV, DENV1, and DENV3, in that order.

### Statistical analysis

The primary outcome of the study was seropositivity to YFV, DENV, CHIKV, and WNV, measured as the presence of IgA/IgM/IgG antibodies. Seropositivity rates for each virus, as well as overall seropositivity rates, were calculated. Descriptive statistics were used to summarize socio-demographic, clinical, and environmental variables.

Socio-demographic, clinical, and environmental factors were assessed for their association with arbovirus seropositivity. Predictor variables included age, sex, school attendance, vaccination status, caregiver characteristics, and behavioural, clinical, and environmental factors.

Since this was a cross-sectional study, we used Prevalence Ratio (PR) to conservatively estimate the strength of the association between different factors and arbovirus seroprevalence. PR and the associated 95% confidence intervals (CIs) were determined using Generalized Linear Model (glm) with the link (probit) function in Stata. Variables with a $p < 0.20$ in univariate analysis (Un-adjusted PRs - UPR) were included in the multivariate analysis model, and adjusted prevalence ratios (aPR) for each arbovirus determined. All statistical analyses were performed using Stata/SE version 17.0 for Mac (StataCorp, College Station, Texas, USA), with statistical significance set at $p \leq 0.05$.

### Ethical statement

This study was approved by the ethical review committee of Kenyatta National Hospital – University of Nairobi Ethics and Research Committee (P108/03/2010). Informed written consent was obtained from all caregivers, and verbal assent was obtained from children above seven years of age before participating in the study. Written parental consent was obtained for all children below two years. The study was conducted in line with the requirements of the Helsinki Declaration of 1975, as revised in 2000.

## Results

A total of 656 children, consisting of 316 (48.2%) males and 340 (51.8%) females, were successfully recruited. Socio-demographic, clinical and environmental data were collected, and all were screened for YFV and DENV serotype 2. Due to insufficient sample volumes, varying numbers of participants were screened for WNV (n = 649), CHIKV (n = 649), DENV serotype 1 (n = 368), and DENV serotype 3 (n = 203). Participant characteristics have been previously reported [33] and are summarized below (Table 1). The complete dataset is available as S1 Table.

**Table 1. Demographic features and frequency of exposures of the study population to anti-arbovirus IgA/IgM/IgG Sero complex antibodies.**

| Participant characteristics | Total population n (%) | Any arbovirus positive n (%) | YFV Positive n (%) | Any DENV Positive n (%) | CHIK positive n (%) | WNV positive n (%) |
|---|---|---|---|---|---|---|
| Gender (n=656) | | | | | | |
| **Male** | 316 (48.2) | 74 (23.4) | 11 (3.48) | 35 (11.1) | 14 (4.5) | 29 (9.3) |
| **Female** | 340 (51.8) | **108 (31.8)**[c] | 18 (5.29) | **68 (20.0)**[c] | 22 (6.5) | 33 (9.8) |
| Age group (Years) (n=654) | | | | | | |
| **1 - 3** | 267 (40.8%) | 54 (20.2) | 16 (5.99) | 35 (13.1) | 10 (3.8) | 10 (3.8) |
| **>3 - 6** | 202 (30.9%) | 57 (28.2) | 10 (4.95) | 24 (11.9) | 11 (5.5) | 27 (13.4) |
| **>6 - 9** | 92 (14.1%) | 31 (33.7) | 2 (2.17) | **23 (25.0)**[d] | 4 (4.5) | 11 (12.4) |
| **>9 - 12** | 93 (14.2%) | **39 (41.9)**[d] | 1 (1.08) | 20 (21.5) | **11 (12.2)**[d] | **14 (15.6)**[d] |
| KEPI Vaccinated (n=656) | | | | | | |
| **No** | 14 (2.1) | 6 (42.9) | 0 (0.0) | 3 (21.4) | 1 (8.3) | **4 (33.3)**[e] |
| **Yes** | 642 (97.9) | 176 (27.4) | 29 (4.5) | 100 (15.6) | 35 (5.5) | 58 (9.1) |
| KEPI Vaccines completed (n=643) | | | | | | |
| **No** | 92 (14.3) | 27 (29.3) | 2 (2.17) | 13 (14.1) | **10 (10.9)**[f] | 9 (9.8) |
| **Yes** | 551 (85.7) | 148 (26.9) | 27 (4.9) | 85 (15.4) | 23 (4.2) | 52 (9.4) |
| Yellow fever vaccinated (n=655) | | | | | | |
| **No** | 644 (98.3) | 181 (28.1) | 29 (4.50) | 103 (16.0) | 36 (5.7) | 61 (9.6) |
| **Yes** | 11 (1.7) | 1 (9.1) | 0 (0.0) | 0 (0.0) | 0 (0.0) | 1 (9.1) |
| School Attendance (n=656) | | | | | | |
| **None** | 347 (52.9) | 81 (23.3) | 19 (5.5) | 46 (13.3) | 16 (4.6) | 24 (6.9) |
| **In school** | 309 (47.1) | 101 32.7)[g] | 10 (3.2) | 57 (18.5) | 20 (6.6) | **38 (12.6)**[g] |
| Usual Residence (n=656) | | | | | | |
| **Town/urban** | 115 (17.5) | 33 (28.7) | 4 (3.5) | 16 (13.9) | 5 (4.4) | **19 (16.7)**[h] |
| **Village/Rural** | 532 (81.1) | 148 (27.8) | 25 (4.7) | 87 (16.4) | 30 (5.7) | 43 (8.2) |
| **Both rural and urban** | 9 (1.4) | 1 (11.1) | 0 (0.0) | 0 (0.0) | 1 (11.1) | 0 (0.0) |
| Primary Care giver (n=655) | | | | | | |
| **Parent** | 590 (90.1) | 151 (25.6) | 25 (4.2) | 88 (14.9) | 29 (5.0) | 50 (8.9) |
| **Grand parent** | 39 (5.9) | 14 (35.9) | 3 (7.7) | 7 (18.0) | 3 (8.3) | 4 (11.1) |
| **Other** | 26 (4.0) | **16 (61.5)**[i] | 1 (3.9) | 7 (26.9) | **4 (15.4)**[i] | **7 (26.9)**[i] |

[c]Significant difference between the sexes (p<0.01) by chi square test.

[d]Significant difference between the age groups (p<0.01) by chi square test

[e]Significant difference between those who did not get any routine vaccinations (p=0.005) by chi square test

[f]Significant difference between those who did not complete routine vaccinations (p=0.005) by chi square test

[g]Significant difference between school attendance and none (p=0.014) by chi square test

[h]Significant difference between residences (p=0.012) by chi square test

[i]Significant difference between the primary caregivers (p<0.05) by chi square test

## Factors associated with seropositivity to any arbovirus

Univariate and multivariate analyses of the selected factors associated with seropositivity to any arbovirus are presented in Table 2. Among the study participants, 182 (27.7%) were seropositive for at least one of the four arboviruses. Female participants had a significantly higher seroprevalence compared to males (UPR 1.29, 95% CI 1.05–1.58, p=0.017); this difference was also seen in the multivariate model (aPR 1.27, 95% CI 1.02–1.59, p=0.035).

In the univariate model, children aged 6–9 years (UPR 1.63, 95% CI 1.19–2.24, p=0.002) and 9–12 years (UPR 1.8, 95% 1.29–2.51, p=0.001) showed significantly higher seroprevalence compared to the reference group aged 1–3 years.

**Table 2. Univariate and multivariate regression analysis of selected factors for Any Arbovirus seropositivity.**

| Characteristic | Number (%) | Univariate Analysis | | | Multivariate Analysis | | |
|---|---|---|---|---|---|---|---|
| | | UPR | 95% CI | p-value | aPR | 95% CI | p-value |
| **Sex** | 656 | | | | | | |
| Male | 316(48.17) | Ref | – | | | | |
| Female | 340(51.83) | 1.29 | 1.05, 1.58 | 0.017** | 1.27 | 1.02, 1.59 | 0.035** |
| **Age Category** | 654 | | | | | | |
| 1-3y | 228(34.86) | Ref | – | | | | |
| 3-6y | 211(32.26) | 1.25 | 0.96, 1.63 | 0.091 | 1.13 | 0.85, 1.49 | 0.402 |
| 6-9y | 99(15.14) | 1.63 | 1.19, 2.24 | 0.002** | 1.43 | 1.02, 2.01 | 0.037** |
| 9-12y | 82(12.54) | 1.8 | 1.29, 2.51 | 0.001** | 1.48 | 1.01, 2.16 | 0.042** |
| **Kepi card completed** | 651 | | | | | | |
| No | 132 (20.28) | Ref | | | | | |
| Yes | 519 (79.72) | 0.77 | 0.60, 0.98 | 0.037** | 1.03 | 0.77, 1.37 | 0.833 |
| **Primary Carer** | 655 | | | | | | |
| Parent | 590(90.08) | Ref | – | | | | |
| Grandparent | 39(5.95) | 1.34 | 0.88, 2.04 | 0.167 | 1.22 | 0.76, 1.95 | 0.411 |
| Other | 26(3.97) | 2.58 | 1.56, 4.27 | 0.000** | 1.94 | 1.11, 3.38 | 0.020** |
| **Mosquito Control Measures** | 655 | | | | | | |
| None | 512(78.17) | Ref | – | | | | |
| Repellents | 73(11.15) | 1.25 | 0.91, 1.72 | 0.164 | 1.16 | 0.81,1.65 | 0.413 |
| Sprays | 70(10.69) | 1.16 | 0.84, 1.61 | 0.376 | 0.99 | 0.68,1.44 | 0.957 |
| **Type of Bed net** | 656 | | | | | | |
| Permanet | 290(44.21) | Ref | | | | | |
| Mixed | 51(7.77) | 1.16 | 0.77, 1.74 | 0.482 | 1.04 | 0.68, 1.59 | 0.872 |
| None | 14(2.13) | 1.55 | 0.78, 3.10 | 0.214 | 1.45 | 0.65, 3.25 | 0.360 |
| Olyset | 211(31.16) | 1.62 | 1.17, 1.89 | 0.001** | 1.28 | 0.99, 1.67 | 0.063 |
| Supanet | 67(10.21) | 1.21 | 0.84, 1.73 | 0.306 | 0.98 | 0.64, 1.51 | 0.941 |
| Unknown | 23(3.51) | 2.64 | 1.54, 4.53 | 0.000** | 2.28 | 1.24, 4.17 | 0.008** |
| **Type of Roof** | 656 | | | | | | |
| iron sheets | 387(58.99) | Ref | – | | | | |
| grass | 228(34.76) | 0.86 | 0.69, 1.07 | 0.176 | 0.87 | 0.68,1.11 | 0.258 |
| iron sheets/tiles | 41(6.25) | 1.41 | 0.94, 2.13 | 0.096 | 1.22 | 0.77, 1.93 | 0.386 |
| **Feeling Sick** | 656 | | | | | | |
| No | 379(57.77) | Ref | – | | | | |
| Yes | 277(42.23) | 0.8 | 0.65, 0.99 | 0.036** | 0.95 | 0.74,1.22 | 0.668 |
| **Rash** | 656 | | | | | | |
| No | 451(68.75) | Ref | – | | | | |
| Yes | 205(31.25) | 1.26 | 1.01, 1.57 | 0.038** | 1.07 | 0.79,1.45 | 0.658 |
| **Past Rash** | 656 | | | | | | |
| **No** | 563(85.82) | Ref | – | | | | |
| **Yes** | 93(14.18) | 1.21 | 0.91, 1.61 | 0.199 | 0.93 | 0.63,1.38 | 0.738 |

Abbreviations: CI = Confidence Interval, UPR = Unadjusted Prevalence Ratio, aPR = Adjusted Prevalence Ratio, ** = p < 0.05

In the multivariate model, children aged 6–9 years had an increased rate of seropositivity (aPR 1.43, 95% CI 1.02–2.01, p = 0.037), while those aged 9–12 years had an aPR of 1.48 (95% CI 1.01–2.16, p = 0.042) when compared to children aged 1–3 years.

Children with "Other" caregivers had a higher rate of seropositivity compared to those with parental caregivers (UPR 2.58, 95% CI 1.56–4.27, p < 0.001) in univariate regression; this finding was replicated in the multivariate model (aPR 1.94, 95% CI 1.11–3.38, p = 0.020).

The rate of seropositivity was also higher in children using Olyset (UPR 1.62, 95% CI 1.17–1.89, *p* = 0.001) or unbranded/unknown bed nets (UPR 2.64, 95% CI 1.54–4.53, *p* < 0.001) compared to those using Permanet nets. The association between unbranded nets and seroprevalence was replicated in the multivariate model (aPR 1.24, 95% CI 1.24–4.17, p = 0.008).

Presenting with a completed KEPI card was associated with lower seroprevalence to any arbovirus (UPR 0.77, 95% CI 0.60–0.98, p = 0.037. So also, was presenting with "feeling sick" (UPR 0.8, 95% CI 0.65–0.99, p = 0.036). None of these was replicated in multivariate analysis.

Finally, presenting with a rash was associated with higher seroprevalence (UPR 1.26, 95% CI 1.06–1.57, p = 0.038). This was not seen in the multivariate model.

### Factors associated with Seropositivity to Yellow Fever, Dengue, Chikungunya and West Nile Viruses

Factors associated with seropositivity to YFV, DENV, CHIKV and WNV are presented in Tables 3–6 respectively.

**Yellow Fever Virus.** No positive or negative associations were noted either in the univariate or multivariate models of factors for YFV seroprevalence (Table 3).

**Dengue virus.** Females had a higher rate of seropositivity than males to any of the three DENV serotypes (UPR 1.46, 95% CI 1.15–1.86, p = 0.002) in univariate analysis; this was confirmed in the multivariate model (aPR 1.62, 95% CI 1.24–2.10, p < 0.001). Dengue antibody prevalence was higher in children aged 6–9 years (UPR 1.54, 95% CI 1.10–2.16, p = 0.012) compared to those aged 1–3 years; this was also seen in the multivariate model (aPR 1.80, 95% CI 1.23–2.64,

**Table 3. Univariate and multivariate logistic regression for YFV seroprevalence.**

| Characteristic | Number (%) | Univariate Analysis | | | Multivariate Analysis | | |
|---|---|---|---|---|---|---|---|
| | | UPR | 95% CI | p-value | aPR | 95% CI | p-value |
| **Age Category** | 654 | | | | | | |
| 1-3y | 228(34.86) | Ref | – | | | | |
| 3-6y | 211(32.26) | 0.87 | 0.58, 1.30 | 0.491 | 1.02 | 0.64, 1.63 | 0.936 |
| 6-9y | 99(15.14) | 1.03 | 0.64, 1.65 | 0.899 | 1.51 | 0.72, 3.17 | 0.271 |
| 9-12y | 82(12.54) | 0.51 | 0.23, 1.13 | 0.098 | 0.76 | 0.29, 2.01 | 0.578 |
| **Attend school** | 656 | | | | | | |
| No | 347(52.90) | Ref | | | | | |
| Yes | 309(47.10) | 0.78 | 0.55, 1.11 | 0.164 | 0.87 | 0.49, 1.54 | 0.630 |
| **Kepi card completed** | 651 | | | | | | |
| No | 132(20.28) | Ref | | | | | |
| Yes | 519(79.72) | 1.43 | 0.86, 2.37 | 0.167 | 1.45 | 0.86, 2.44 | 0.161 |
| **Waterbodies** | 656 | | | | | | |
| **No** | 385(58.69) | Ref | | | | | |
| **Yes** | 271(41.31) | 0.69 | 0.48, 1.01 | 0.055 | 0.69 | 0.47, 1.03 | 0.071 |

Abbreviations: CI = Confidence Interval, UPR = Unadjusted Prevalence Ratio, aPR = Adjusted Prevalence Ratio, ** = p < 0.05

**Table 4. Univariate and Multivariate Logistic Regression for DENV seroprevalence.**

| Characteristic | Number (%) | Univariate Analysis | | | Multivariate Analysis | | |
|---|---|---|---|---|---|---|---|
| | | UPR | 95% CI | p-value | aPR | 95% CI | p-value |
| **Sex** | 656 | | | | | | |
| Male | 316(48.17) | Ref | – | | | | |
| Female | 340(51.83) | 1.46 | 1.15, 1.86 | 0.002** | 1.62 | 1.24, 2.10 | 0.000** |
| **Age Category** | 654 | | | | | | |
| 1-3y | 228(34.86) | Ref | – | | | | |
| 3-6y | 211(32.26) | 0.85 | 0.63, 1.16 | 0.309 | 0.91 | 0.65,1.27 | 0.592 |
| 6-9y | 99(15.14) | 1.54 | 1.10, 2.16 | 0.012** | 1.8 | 1.23, 2.64 | 0.003** |
| 9-12y | 82(12.54) | 1.38 | 0.96, 1.99 | 0.082 | 1.54 | 1.00, 2.38 | 0.050** |
| **KEPI card completed** | 651 | | | | | | |
| No | 132(20.28) | Ref | | | | | |
| Yes | 519(79.72) | 0.69 | 0.53, 0.90 | 0.007** | 0.95 | 0.68, 1.32 | 0.747 |
| **Primary Carer** | 655 | | | | | | |
| Parent | 590(90.08) | Ref | – | | | | |
| Grandparent | 39(5.95) | 1.13 | 0.70, 1.82 | 0.614 | 1.23 | 0.73, 2.09 | 0.428 |
| Other | 26(3.97) | 1.53 | 0.90, 2.60 | 0.117 | 1.48 | 0.77, 2.84 | 0.237 |
| **Bednet brand** | 656 | | | | | | |
| Permanet | 290(44.21) | Ref | | | | | |
| Mixed | 51(7.77) | 0.83 | 0.51, 1.34 | 0.447 | 0.69 | 0.41, 1.15 | 0.150 |
| None | 14(2.13) | 1 | – | – | 1 | – | – |
| Olyset | 211(31.16) | 1.15 | 0.88, 1.49 | 0.299 | 1.08 | 0.80, 1.46 | 0.628 |
| Supanet | 67(10.21) | 0.71 | 0.45, 1.12 | 0.141 | 0.78 | 0.43, 1.41 | 0.407 |
| Unknown | 23(3.51) | 1.06 | 0.57, 1.99 | 0.848 | 0.95 | 0.46, 1.96 | 0.899 |
| **Type of Roof** | 656 | | | | | | |
| Iron sheets | 387(58.99) | Ref | – | | | | |
| Grass thatch | 228(34.76) | 0.77 | 0.59, 1.00 | 0.047** | 0.84 | 0.63, 1.12 | 0.228 |
| Iron sheets/tiles | 41(6.25) | 0.97 | 0.60, 1.56 | 0.904 | 1.12 | 0.65, 1.93 | 0.672 |
| **Eaves** | 656 | | | | | | |
| No | 611(93.14) | Ref | | | | | |
| Yes | 45(6.86) | 0.48 | 0.25, 0.93 | 0.029** | 0.68 | 0.33, 1.40 | 0.295 |
| **Waterbodies** | 656 | | | | | | |
| **No** | 385(58.69) | Ref | | | | | |
| **Yes** | 271(41.31) | 0.77 | 0.61, 0.98 | 0.037** | 0.79 | 0.58, 1.07 | 0.127 |
| **Past Rash** | 656 | | | | | | |
| No | 563(85.82) | Ref | – | | | | |
| Yes | 93(14.18) | 0.77 | 0.53, 1.10 | 0.150 | 0.68 | 0.44, 1.06 | 0.088 |
| **Sore throat** | 656 | | | | | | |
| No | 611(93.14) | Ref | | | | | |
| Yes | 45(6.86) | 0.59 | 0.33, 1.06 | 0.076 | 0.41 | 0.22, 0.79 | 0.008** |

Abbreviations: CI = Confidence Interval, UPR = Unadjusted Prevalence Ratio, aPR = Adjusted Prevalence Ratio, ** = p < 0.05

p = 0.03). Although no association was noted in the univariate model DENV antibody seroprevalence was higher in children aged 9–12 years compared to those aged 1–3 years (aPR 1.54, 95% CI 1.00–2.38, p = 0.050) in the multivariate model. (Table 4).

**Table 5. Univariate and Multivariate Logistic Regression for CHIKV seroprevalence.**

| Characteristic | Number (%) | Univariate Analysis | | | Multivariate Analysis | | |
|---|---|---|---|---|---|---|---|
| | | PR | 95% CI | p-value | PR | 95% CI | p-value |
| **Class** | 656 | | | | | | |
| None | 346(52.74) | Ref | | | | | |
| Preschool | 136(20.73) | 1.06 | 0.69, 1.61 | 0.797 | 1.03 | 0.64, 1.67 | 0.886 |
| Lower Primary | 97(14.79) | 1.08 | 0.67, 1.74 | 0.765 | 1.00 | 0.57, 1.77 | 0.937 |
| Upper primary | 76(11.59) | 1.56 | 1.00, 2.44 | 0.049** | 1.57 | 0.95, 2.57 | 0.076 |
| **KEPI card completed** | 651 | | | | | | |
| No | 132(20.28 | Ref | | | | | |
| Yes | 519(79.72) | 0.76 | 0.53, 1.10 | 0.150 | 1.43 | 0.70, 2.90 | 0.328 |
| **Kepi vaccination completed** | 643 | | | | | | |
| No | 92(14.31) | Ref | | | | | |
| Yes | 551(85.69) | 0.61 | 0.41, 0.90 | 0.012** | 0.56 | 0.28, 1.12 | 0.102 |
| **Primary Carer** | 655 | | | | | | |
| Parent | 590(90.08) | Ref | | | | | |
| Grandparent | 39(5.95) | 1.31 | 0.71, 2.41 | 0.394 | 0.94 | 0.46, 1.92 | 0.866 |
| Other | 26(3.97) | 1.88 | 1.02, 3.45 | 0.043** | 1.57 | 0.84, 2.95 | 0.155 |
| **Mosquito control** | 555 | | | | | | |
| None | 512(78.17) | Ref | | | | | |
| Repellents | 73(11.15) | 2.03 | 1.36, 3.04 | 0.001** | 2.54 | 1.62, 3.98 | 0.000** |
| Sprays | 70(10.69) | 1.17 | 0.70, 1.95 | 0.556 | 1.09 | 0.57, 2.11 | 0.793 |
| **Type of Bednet** | 656 | Ref | | | | | |
| Permanet | 290(44.21) | | | | | | |
| Mixed | 51(7.77) | 1.02 | 0.51, 2.01 | 0.965 | 1.13 | 0.54, 2.37 | 0.751 |
| None | 14(2.13) | 2.67 | 1.22, 5.86 | 0.014** | 2.5 | 1.03, 6.06 | 0.042** |
| Olyset | 211(31.16) | 1.34 | 0.92, 1.94 | 0.130 | 1.28 | 0.84, 1.93 | 0.248 |
| Supanet | 67(10.21) | 0.9 | 0.46, 1.73 | 0.748 | 0.87 | 0.43, 1.75 | 0.688 |
| Unknown | 23(3.51) | 2.31 | 1.19, 4.47 | 0.013** | 1.9 | 0.92, 3.93 | 0.082 |
| **Type of Roof** | 656 | | | | | | |
| Iron sheets | 387(58.99) | Ref | | | | | |
| Grass thatch | 228(34.76) | 1.38 | 0.99, 1.91 | 0.054 | 1.59 | 1.11, 2.28 | 0.012** |
| Iron sheets/tiles | 41(6.25) | 1.08 | 0.54, 2.14 | 0.837 | 0.46 | 0.18, 1.18 | 0.106 |
| **Waterbodies** | 656 | | | | | | |
| No | 385(58.69) | Ref | | | | | |
| Yes | 271(41.31) | 1.32 | 0.96, 1.82 | 0.084 | 0.71 | 0.45, 1.11 | 0.131 |
| **Feeling Sick** | 656 | | | | | | |
| No | 379(57.77) | | | | | | |
| Yes | 277(42.23) | 0.64 | 0.45, 0.91 | 0.014** | 0.7 | 0.46, 1.08 | 0.109 |
| **Past rash** | 656 | | | | | | |
| No | 451(68.75) | | | | | | |
| Yes | 205(31.25) | 1.55 | 1.05, 2.28 | 0.026** | 1.43 | 0.89, 2.28 | 0.138 |

Abbreviations: CI=Confidence Interval, UPR=Unadjusted Prevalence Ratio, aPR=Adjusted Prevalence Ratio, **=p<0.05

**Table 6. Univariate and Multivariate Logistic Regression for WNV seroprevalence.**

| Characteristic | Number (%) | Univariate Analysis | | | Multivariate Analysis | | |
|---|---|---|---|---|---|---|---|
| | | PR | 95% CI | p-value | PR | 95% CI | p-value |
| **Age Category** | 654 | | | | | | |
| 1-3y | 228(34.86) | Ref | | | | | |
| 3-6y | 211(32.26) | 1.94 | 1.31, 2.88 | 0.001** | 2.14 | 1.35, 3.40 | 0.001** |
| 6-9y | 99(15.14) | 2.28 | 1.46, 3.56 | 0.000** | 2.9 | 1.56, 5.40 | 0.001** |
| 9-12y | 82(12.54) | 2.18 | 1.36, 3.49 | 0.001** | 2.27 | 1.19, 4.34 | 0.013** |
| **Residence** | 656 | | | | | | |
| Village | 115(17.53) | Ref | | | | | |
| Town | 532(81.10) | 1.53 | 1.12, 2.10 | 0.005** | 1.50 | 1.02, 2.20 | 0.040** |
| Other | 9(1.37) | 1.00 | – | – | 1.00 | – | – |
| **Attend school** | 656 | | | | | | |
| No | 347(52.90) | Ref | | | | | |
| Yes | 309(47.10) | 1.40 | 1.07, 1.83 | 0.015** | 0.70 | 0.46, 1.09 | 0.113 |
| **Primary Carer** | 655 | | | | | | |
| Parent | 590(90.08) | Ref | | | | | |
| Grandparent | 39(5.95) | 1.16 | 0.66, 2.04 | 0.600 | 0.75 | 0.38, 1.49 | 0.408 |
| Other | 26(3.97) | 2.13 | 1.24, 3.64 | 0.006** | 1.67 | 0.85, 3.28 | 0.136 |
| **Type of Bednet** | 656 | | | | | | |
| Permanet | 290(44.21) | Ref | | | | | |
| Mixed | 51(7.77) | 1.75 | 0.97, 3.16 | 0.060 | 2.34 | 1.22, 4.50 | 0.010** |
| None | 14(2.13) | 3.27 | 1.47, 7.27 | 0.004** | 2.42 | 0.74, 7.96 | 0.145 |
| Olyset | 211(31.16) | 2.52 | 1.73, 3.68 | 0.000** | 2.75 | 1.73, 4.35 | 0.000** |
| Supanet | 67(10.21) | 2.55 | 1.57, 4.12 | 0.000** | 2.18 | 1.25, 3.80 | 0.006** |
| Unknown | 23(3.51) | 4.87 | 2.64, 8.99 | 0.000** | 5.06 | 2.46, 10.41 | 0.000** |
| **Type of Roof** | 656 | | | | | | |
| Iron sheets | 387(58.99) | Ref | | | | | |
| grass | 228(34.76) | 1.12 | 0.83, 1.50 | 0.460 | 1.34 | 0.94, 1.92 | 0.110 |
| Iron sheets/tiles | 41(6.25) | 2.43 | 1.55, 3.79 | 0.000** | 1.96 | 1.20, 3.21 | 0.007** |
| **Water bodies near house** | 656 | | | | | | |
| No | 385 (58.68) | Ref | | | | | |
| Yes | 271(41.31) | 1.69 | 1.29, 2.23 | 0.000** | 0.95 | 0.64, 1.40 | 0.779 |
| **Vegetation around the house** | 656 | | | | | | |
| No | 411(62.65) | Ref | | | | | |
| Yes | 245(37.35) | 1.34 | 1.03, 1.76 | 0.032** | 1.08 | 0.75, 1.56 | 0.681 |
| **Feeling sick** | 656 | | | | | | |
| No | 379(57.77) | Ref | | | | | |
| Yes | 277(42.23) | 0.67 | 0.50, 0.90 | 0.007** | 1.16 | 0.78, 1.72 | 0.466 |
| **Rash** | 656 | | | | | | |
| No | 451(68.75) | Ref | | | | | |
| Yes | 205(31.25) | 1.69 | 1.29, 2.23 | 0.000** | 1.31 | 0.83, 2.06 | 0.242 |
| **Past rash** | 656 | | | | | | |
| **No** | 451(68.75) | Ref | | | | | |
| **Yes** | 205(31.25) | 1.64 | 1.17, 2.29 | 0.004** | 1.24 | 0.77, 1.98 | 0.375 |
| **Sore throat** | 656 | | | | | | |
| No | 563(85.82) | Ref | | | | | |
| Yes | 93(14.18) | 1.38 | 0.87, 2.20 | 0.180 | 1.09 | 0.64, 1.83 | 0.757 |

Abbreviations: CI = Confidence Interval, UPR = Unadjusted Prevalence Ratio, aPR = Adjusted Prevalence Ratio, ** = p < 0.05

Presenting with a fully completed KEPI card (UPR 0.69, 95% CI 0.53–0.90, p = 0.007), living in a grass thatched house as opposed to an iron sheets-roofed house (UPR 0.77, 95% CI 0.59–1.00, p = 0.047), living in a house that had eaves (UPR 0.48, 95% CI 0.25–0.93, p = 0.029), and living near water bodies (UPR 0.77, 95% CI 0.61–0.98, p = 0.037) were all associated with lower DENV seroprevalence, but this effect was not replicated in the multivariate model (p > 0.05).

**Chikungunya Virus.** For CHIKV, those in "Upper Primary School" had a significantly higher seroprevalence in univariate analysis (UPR 1.56, 95% CI 1.00–2.44, p = 0.049); in the multivariate model, this was not seen. Children whose caregiver was "Other" had higher CHIKV seroprevalence compared to those with parental caregivers (UPR 1.8 95% CI 1.02–3.45, p = 0.043); this association was not seen in multivariate analysis (p > 0.05). (Table 5).

Children who lived in homes that used mosquito repellents had a higher seroprevalence of CHIKV antibodies when compared with those who did not (UPR 2.03, 95% CI 1.36–3.04, p = 0.001); this was replicated in the multivariate model (aPR 2.54, 95% CI 1.62–3.98, p < 0.001).

Those who did not sleep under any bed net had a higher CHIKV seroprevalence (UPR 2.67, 95% CI 1.22–5.86, p = 0.014) when compared to those who used the brand "Permanet". In multivariate analysis, this association was also seen (aPR 2.5, 95% CI 1.03–6.06, p = 0.042). Moreover, even those who used an unknown brand of net had higher seropositivity (UPR 2.31, 95% CI 1.19–4.47, p = 0.013) when compared to those who used the brand "Permanet", but this was not seen in the multivariate model (p > 0.05). Those who lived in houses thatched with grass had a higher prevalence of CHIKV antibodies in multivariate analysis (aPR 1.59, 95% CI 1.11–2.28, p = 0.012), although this was not seen in the univariate analysis.

Children who reported a past rash showed high CHIKV seroprevalence (UPR 1.55, 95% CI 1.05–2.28, p = 0.026) compared to those who did not; this was not seen during the multivariate analysis.

Those who had completed their KEPI vaccination schedule (UPR 0.61, 95% CI 0.41–0.90, p = 0.012), and those who presented feeling sick (UPR 0.64, 95% CI 0.45–0.91, p = 0.014), all showed lower CHIKV seroprevalence in univariate but not multivariate analysis.

**West Nile Virus.** For WNV, antibody seroprevalence in those aged 3–6 (PR 1.94, 95% CI 1.31–2.88, p = 0.001), 6–9 (UPR 2.28, 95% CI 1.46–3.56, p < 0.001) and 9–12 (UPR 2.18, 95% CI 1.36–3.49, p = 0.001) year age groups were significantly higher compared to those aged 1–3 years. In the multivariate model the aPR was also higher for all age groups when compared to 1–3 years: in those aged 3–6 it was 2.14 (95% 1.35–3.40, p = 0.001), in those aged 6–9 it was 2.9 (95% CI 1.56–5.40, p = 0.001), while for those aged 9–12 years it was 2.27 (95% CI 1.19–4.34, p = 0.013). (Table 6).

Living in a town/urban area was associated with a higher seroprevalence of WNV (UPR 1.53, 95% CI 1.12–2.10, p = 0.005) when compared with living in the village/rural area, and this was replicated in the multivariate model (aPR 1.50, 95% CI 1.02–2.20, p = 0.040). Attending school was also associated with higher seroprevalence (UPR 1.40, 95% CI 1.07–1.83, p = 0.015) but only in the univariate model.

Those children whose caregiver was "Other" had a higher seroprevalence of WNV antibodies compared to those with parental caregivers (UPR 2.13, 95% CI 1.24–3.64, p = 0.006), but this was not replicated in the multivariate model.

WNV seroprevalence was also higher in children using no mosquito bed nets (UPR 3.27, 95% CI 1.47–7.27, p = 0.004), using "Olyset" (PR 2.52, 95% CI 1.73–3.68, p < 0.001), "Supanet" (UPR 2.55, 95% CI 1.57–4.12, p = 0.001), or unknown brands of bed nets (UPR 4.87, 95% CI 2.64–8.99, p < 0.001) when compared to those using "Permanet" bed nets in univariate analysis. These associations remained significant in the multivariate model (p < 0.05) except for those using no nets at all (aPR 2.42 95% CI 0.74–7.96, p = 0.145).

Those living in houses that were roofed with a mix of iron sheets and tiles had higher WNV seroprevalence (UPR 2.43, 95% CI 1.55–3.79, p < 0.001) compared to those living in houses roofed with iron sheets only. This was affirmed in the multivariate model (aPR 1.96, 95% CI 1.20–3.21, 0.007).

Having water bodies near the house (UPR 1.69, 95% CI 1.29–2.23, p < 0.001), vegetation around the house (UPR 1.34, 95% CI 1.03–1.76, p = 0.032), an active rash (UPR 1.69, 95% CI 1.29–2.23, p = 0.001), or a past rash (UPR 1.64, 95%

CI 1.17–2.29, p = 0.004) were all associated with higher seroprevalence of WNV but this was not seen in the multivariate model.

Those who presented feeling sick (UPR 0.67, 95% CI 0.50–0.90, p = 0.007) all had lower WNV seroprevalence in univariate but not in multivariate analysis.

## Discussion

In this cross-sectional study conducted in Western Kenya, we identified several factors: age, female gender, non-parental primary caregiver, urban residence, mosquito net usage, bed net type, use of insect repellents, and house roofing material, that are significantly associated with exposure to arboviral infections.

Our findings suggest that females exhibited a higher rate of seropositivity to dengue virus (DENV). This finding has been demonstrated in other studies. This could be attributed to behavioural and environmental factors. *Aedes aegypti*, the primary vector for DENV, predominantly resides and bites in outdoor environments in daytime, such as in garden vegetation, but can also reside and bite in indoor environments, such as kitchens and bedrooms. The mosquito bites exposed areas such as the lower legs, ankles, and arms. In Western Kenya, women are often more engaged in outdoor activities such as gardening, fetching water or firewood, and indoor chores such as cooking, which puts them at greater risk of mosquito bites compared to men. Additionally, they are less likely to wear protective clothing while performing these activities. Similar patterns have been observed in other sub-Saharan African studies, where women's involvement in both domestic and outdoor activities increase their exposure to daytime-biting mosquitoes [23,27–29].

In this study, seropositivity for any arbovirus increased with age, similar to other studies that have found an association between increasing age and arbovirus exposure [11,34]. Children aged 3–9, 6–9 and 9–12 years had a higher rate of seropositivity for WNV, while those aged 6–9 years showed significantly higher seropositivity to DENV compared to younger children (1–3 years). *Culex* mosquitoes, which transmit WNV, breed in stagnant water near human habitation, and are most active and bite humans primarily at dusk and dawn. They are also known to bite throughout the night. In Western Kenya, older children often play outdoors, particularly during these peak *Culex* mosquito activity periods, and frequently interact with water sources, such as stagnant pools, broken pots and discarded containers and used tires. School attendance, which involves outdoor activities during these periods, likely contributes to the higher exposure to both *Aedes* mosquitoes during the day and *Culex* mosquitoes at dawn and dusk [30–33].

Children whose primary caregivers were not their parents or grandparents had a higher seropositivity to any arbovirus, likely reflecting the protective roles that parents and grandparents play in limiting outdoor exposure to mosquitoes and ensuring children wear appropriate clothing.

We also observed a higher prevalence of WNV in individuals residing in homes with iron sheet and tile roofs. Water collected from iron sheet and tile roofs in rainwater barrels, catch basins, storm drains and others provide relatively clean water, an ideal habitat for breeding *Culex* mosquitoes, with the tiles also offering a shaded and humid environment that fosters the development of mosquito larvae in the small water pools that form between them. Moreover, the tiles serve as shelter for adult mosquitoes during their less active daytime hours. In contrast, grass-thatched roofs were associated with a higher seropositivity to chikungunya virus (CHIKV). This may be due to the use of pots and gourds for storage of water in these homes, which are likely to harbour *Aedes* mosquitoes, as these mosquitoes tend to rest in the inner linings of such containers.

Urban residence was also associated with higher WNV seroprevalence, which can be attributed to the increased abundance of *Culex* mosquitoes in urban areas, where stagnant water from drainage systems, old tires, puddles, and other waste provides breeding sites. This finding agrees with some studies that have correlated urbanization with WNV disease [35,36].

An intriguing finding was that the use of insect repellents was associated with a higher seropositivity to CHIKV compared to using sprays or not using any repellent. This may suggest that some repellents are less effective against *Aedes* mosquitoes, and tend to be used during non-peak *Aedes* mosquito bite times, possibly providing users with a false sense

of protection. Additionally, individuals using repellents may be more likely to reside in thatched-roof homes, which are associated with higher seropositivity to CHIKV, or may have inadequate mosquito protection during daytime hours when *Aedes* mosquitoes predominantly bite.

Interestingly, although mosquito nets are effective in preventing malaria, most brands did not significantly reduce WNV seroprevalence. In fact, when compared to Permanet, all other brands were significantly less effective in reducing WNV exposure, which corroborates previous findings [37]. Given that *Culex* mosquitoes are active at dusk and dawn, mosquito nets may have limited efficacy against them. Regarding CHIKV, the use of unknown brands of mosquito nets was notably detrimental. The reduced effectiveness of mosquito nets could be attributed to factors such as wear and tear or insufficient insecticide treatment, which is often seen in well-established brands, underscoring the need for alternative prevention strategies. In addition, *Aedes Aegypti*, predominantly bite outdoors, during daytime, when bed nets are unlikely to be of help.

Clinical symptoms such as fever and rash were evaluated for their association with seropositivity, but only "feeling sick", and a rash were significantly linked to arbovirus seropositivity in the univariate, but not in the multivariate models: "Feeling Sick" was associated with a decrease in any arbovirus, CHIKV and WNV seropositivity, while a rash was associated with increased risk of any arbovirus, CHIKV and WNV seropositivity. However, the cross-sectional design of our study, and the lack of differentiation between antibody subtypes (hence no differentiation between recent and past arbovirus infections) limits our ability to establish causality, as other conditions with overlapping symptoms, such as malaria and typhoid, could confound these results, and current symptoms could not be attributed to active arbovirus infection.

We detected antibodies in infants and young children aged 1-3y. Whereas maternal IgA and IgG antibodies are transmitted via breastfeeding and may influence seropositivity results in breastfeeding infants, these antibodies decline within a year of cessation of breastfeeding [38]. Nonetheless, our study was not designed to delineate seropositivity induced by mosquito bites from that induced by maternal antibodies. Ergo, we did not attribute the clinical symptoms to ongoing arbovirus infection in this age group.

Some factors were significantly associated with arbovirus seropositivity in the univariate analysis, but were not significant in the multivariate model. For instance, having a completed KEPI card, reporting "feeling sick", having a rash and olyset bed net were associated with any arbovirus; completed KEPI card, eaves, and water bodies near the house with DENV; attending upper primary school, completed vaccinations, "Other" carer givers, unknown bed nets, feeling sick and a past rash with CHIKV; while attending school, vegetation and water bodies near the house; "Other" care givers, not using bed nets, feeling sick, an ongoing and a past rash were associated with WNV in the univariate analysis. None of these were replicated in the multivariate analysis. This may be explained by confounding by other variables in the multivariate model.

## Conclusion

Our study highlights the significant role of sociodemographic, behavioural, and environmental factors in determining exposure to arboviral infections in children in Western Kenya. Age, female gender, non-parental primary caregiver, failure to use mosquito bed nets, type of bed net, use of insect repellents, urban residence and house roofing material, were identified as important factors associated with arbovirus exposure among children. These factors are modifiable and may serve as targets for tailored interventions to reduce the burden of arboviral infections. Interventions targeting housing improvements, education on bed nets and mosquito repellent use, and environmental mosquito control measures can help reduce the burden of arbovirus diseases in endemic areas.

## Recommendations

Based on the findings of this study, we propose several interventions to reduce arboviral transmission in Western Kenya:

1. **Gender-Sensitive Prevention Programs**: Programs should be designed to address gender-specific exposure. Women, who are more likely to engage in domestic and outdoor activities that increase exposure to daytime-biting mosquitoes, could benefit from appropriate protective clothing and environmental interventions.

2. **Targeted Mosquito Control**: In addition to traditional mosquito nets, interventions should focus on mosquito repellents effective against daytime-biting mosquitoes and strategies to reduce exposure to *Aedes* mosquitoes, the primary vectors of arboviral transmission and *Culex* mosquitoes.

3. **Enhanced Vector Control in Vegetation-Dense Areas**: Given the association between dense vegetation and increased seropositivity, vector control efforts should be expanded, particularly in areas with abundant vegetation and stagnant water around homes, which provide breeding grounds for mosquitoes.

4. **Further Research on Bed Net Efficacy**: Additional studies are needed to assess the effectiveness of various mosquito net brands in preventing arboviral infections, considering factors such as insecticide treatment, net durability, and the mosquito species involved in transmission. Research should also explore alternative protective measures, including daytime repellents and environmental management of mosquito breeding sites.

By targeting these modifiable factors, it may be possible to reduce the prevalence of arboviral infections and improve public health outcomes in the region.

## Study limitations

The detection of arthropod-borne viruses using ELISA, and interpretation of the results is challenging due to cross reactivity, limiting the confidence in reported outcomes. The ELISA tests employed for this study did not differentiate between the IgA, IgM or IgG immunoglobulin classes. To mitigate this, we employed Plaque Reduction Neutralization Tests (PRNT) to confirm results whenever sample volumes allowed [33]. However, due to the limited sample volumes, we could not estimate the true burden of some of the arboviruses in the study group. Additionally, as this study was hospital-based, there is a potential for selection bias, and the results cannot be extrapolated to the study population. Moreover, the use of stringent ELISA cutoff criteria, as outlined by Igarashi et al., may have excluded some samples that were actually positive. With larger sufficient sample volumes, this limitation could be addressed through PRNT.

## Supporting information

**S1 Table. Complete dataset – Factors for arboviral seropositivity.**
(XLSX)

## Acknowledgments

We want to acknowledge all the people, listed and not listed, whose contributions made this research possible:

Dr Toru Kubo and his team from the Department of Virology (Nagasaki University Institute of Tropical Medicine, Nagasaki, Japan), provided intellectual insights, technical support and supervision.

Research colleagues from the Department of Paediatrics and Child Health (University of Nairobi, Kenya) and Prof. Masaaki Shimada (Nagasaki University Africa Research Station, Nairobi, Kenya), provided insight and expertise that greatly assisted the research.

Joyce Ngoi, Minayo Chahilu, Carolyne Kirwaye, Sheila Kageha and Janet Owando (Kenya Medical Research Institute) provided logistical and technical support and assistance with the laboratory analysis.

Michael Obura, Evans Omao, and the staff of KEMRI Alupe Clinic and Alupe Sub-County Hospital, were instrumental in the participant recruitment, patient assessment and data collection.

Dr Jimmy Wafula and other staff members at Alupe Sub-County Hospital provided the study participants, clinical and administrative support that made the research possible.

We also acknowledge Dr Martin Mulinge for reviewing the manuscript in its second iteration, and Nancy Kagendi for reviewing the regression models.

The parents and guardians, as well as the children who agreed to participate in this study, deserve special thanks.

## Author contributions

**Conceptualization:** Mary Inziani, Kouichi Morita, Matilu Mwau.

**Data curation:** Mary Inziani, Shingo Inoue, Matilu Mwau.

**Formal analysis:** Mary Inziani, Jane Mawia Kilonzo, Marthaclaire Kerubo, Sylvia Mango, Matilu Mwau.

**Funding acquisition:** Kouichi Morita, Matilu Mwau.

**Investigation:** Mary Inziani, Shingo Inoue, Matilu Mwau.

**Methodology:** Mary Inziani, Shingo Inoue, Matilu Mwau.

**Project administration:** Shingo Inoue, Kouichi Morita, Matilu Mwau.

**Resources:** Kouichi Morita, Matilu Mwau.

**Supervision:** Mary Inziani, Shingo Inoue, Kouichi Morita, Matilu Mwau.

**Validation:** Mary Inziani, Jane Mawia Kilonzo, Marthaclaire Kerubo, Sylvia Mango, Allan Ndirangu, Matilu Mwau.

**Visualization:** Mary Inziani, Jane Mawia Kilonzo, Marthaclaire Kerubo, Allan Ndirangu, Matilu Mwau.

**Writing – original draft:** Mary Inziani, Jane Mawia Kilonzo, Marthaclaire Kerubo, Sylvia Mango, Mary Kavurani, Allan Ndirangu, Elizabeth Njeri, Diuniceous Oigara Ogenche, Sylvester Ogolla Ayoro, Shingo Inoue, Kouichi Morita, Matilu Mwau.

**Writing – review & editing:** Mary Inziani, Jane Mawia Kilonzo, Marthaclaire Kerubo, Sylvia Mango, Mary Kavurani, Allan Ndirangu, Elizabeth Njeri, Diuniceous Oigara Ogenche, Sylvester Ogolla Ayoro, Shingo Inoue, Kouichi Morita, Matilu Mwau.

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
