## [Decision Letter · Decision Letter 0]

6 Aug 2025

PONE-D-25-31972Factors for arboviral seropositivity in children in Teso South Sub County, KenyaPLOS ONE

Dear Dr. Inziani,

Thank you for submitting your manuscript to PLOS ONE. After careful consideration, we feel that it has merit but does not fully meet PLOS ONE’s publication criteria as it currently stands. Therefore, we invite you to submit a revised version of the manuscript that addresses the points raised during the review process.

We look forward to receiving your revised manuscript.

Kind regards,

José Ramos-Castañeda, M.Sc., Ph.D

Academic Editor

PLOS ONE

Journal Requirements: 

2. For studies involving third-party data, we encourage authors to share any data specific to their analyses that they can legally distribute. PLOS recognizes, however, that authors may be using third-party data they do not have the rights to share. When third-party data cannot be publicly shared, authors must provide all information necessary for interested researchers to apply to gain access to the data. (https://journals.plos.org/plosone/s/data-availability#loc-acceptable-data-access-restrictions)

4) All necessary contact information others would need to apply to gain access to the data.

Reviewers' comments:

Reviewer's Responses to Questions

**Comments to the Author**

1. Is the manuscript technically sound, and do the data support the conclusions?

Reviewer #1: Partly

Reviewer #2: Yes

2. Has the statistical analysis been performed appropriately and rigorously? 

Reviewer #1: Yes

Reviewer #2: Yes

3. Have the authors made all data underlying the findings in their manuscript fully available?

Reviewer #1: Yes

Reviewer #2: Yes

4. Is the manuscript presented in an intelligible fashion and written in standard English?

Reviewer #1: Yes

Reviewer #2: Yes

5. Review Comments to the Author

Reviewer #1: While this study identifies several important factors associated with arboviral seropositivity in children, the discussion would benefit from acknowledging key limitations. First, the interpretation of IgA/IgM/IgG positivity without confirmatory neutralization tests raises the possibility of cross-reactivity, especially among flaviviruses such as DENV, YFV, and WNV.

The authors should provide more detailed information about the diagnostic assay used to detect arboviral antibodies. Specifically, it would be important to report the commercial brand, manufacturer, type of assay (e.g., ELISA, rapid diagnostic test), and its validated sensitivity and specificity for each virus included (YFV, DENV, CHIKV, WNV). This information is critical to interpret the reliability of seropositivity results and to assess potential cross-reactivity, especially among flaviviruses. Additionally, it would be helpful to specify whether the assay distinguishes between antibody isotypes (IgM, IgG, IgA) and if quantitative results or cutoff thresholds were applied according to manufacturer instructions.

It is important to consider that maternal antibodies transmitted via breastfeeding—especially IgA and, to some extent, IgG—may influence seropositivity results in infants and young children. In such cases, detected antibodies may not reflect active or past infection in the child, but rather passive immunity acquired through breast milk. This possibility should be discussed when interpreting serological findings in pediatric populations.

The study population is limited to children attending healthcare facilities, which may introduce selection bias and limit generalizability to the broader community.

Reviewer #2: Methodology

• Well-structured research design with clear objectives

• Appropriate statistical analysis using both univariate and multivariate models

• Good sample size (656 participants)

• Clear inclusion criteria and ethical considerations

Results Presentation

• Comprehensive data presentation in tables

• Clear breakdown of findings by virus type

• Detailed statistical analysis with confidence intervals and p-values

Discussion

• Thorough interpretation of findings

• Good connection to existing literature

• Clear explanation of potential mechanisms behind observed associations

• Recognition and clear explanation of limitations of study.

Recommendations

• Useful and practical

• Clear view of the knowledge gap, for example studying the effectiveness of different types of net beds

6. PLOS authors have the option to publish the peer review history of their article (what does this mean? ). If published, this will include your full peer review and any attached files.

**Do you want your identity to be public for this peer review?** For information about this choice, including consent withdrawal, please see our Privacy Policy .

Reviewer #1: **Yes: ** Irma Yvonne Amaya-Larios

Reviewer #2: No

---

## [Author Response · Author response to Decision Letter 1]

23 Aug 2025

Response to Reviewers’ - PONE-D-25-31972_Revision2

Dear Editors,

We appreciate your review of our revised manuscript and request. We have addressed your request indicated in italics under each request.

PONE-D-25-31972R1

Factors for arboviral seropositivity in children in Teso South Sub County, Kenya

Dr Mary Inziani

Dear Dr. Inziani,

We've checked your submission and before we can proceed, we need you to address the following issues:

1. In the online submission form, you indicated that [Data cannot be shared publicly because they include clinical data with patient identifiers. Data are available from the Director General, Kenya Medical Research Institute (director@kemri.go.ke) for researchers who meet the criteria for access to confidential data.].

3. Uploaded as supplementary information.

This policy applies to all data except where public deposition would breach compliance with the protocol approved by your research ethics board. If your data cannot be made publicly available for ethical or legal reasons (e.g., public availability would compromise patient privacy), please explain your reasons on resubmission and your exemption request will be escalated for approval."

Response:

Thank you. We have removed identifiers from the study database. We have provided the dataset as supplementary information - Supplementary Table 1 (S1_Table.xlsx).

We have edited the results section and cited the supporting information in the manuscript text, in the Results section, lines 159-167. We have also indicated the Supplementary information at the end of the manuscript (Lines 648-650).

We've returned your manuscript to your account. Please resolve these issues and resubmit your manuscript within 21 days. If you need more time, please email the journal office at plosone@plos.org. We are happy to grant extensions of up to one month past this due date. If we do not hear from you within 21 days, we will withdraw your manuscript.

Response:

We have resolved the issue raised and submitted the revised manuscript.

Please log on to PLOS Editorial Manager at https://www.editorialmanager.com/pone/ to access your manuscript. You will find your manuscript in the 'Submissions Sent Back to Author' link under the New Submissions menu. Be sure to remove your previous manuscript file if you are uploading a new file in response to these requests. After you've made the changes requested above, please be sure to view and approve the revised PDF after rebuilding the PDF to complete the resubmission process.

We are requesting these changes to comply with the PLOS ONE submission guidelines (https://journals.plos.org/plosone/s/submission-guidelines). Please note that we won't send your manuscript for review until you have resolved the above requests.

Thank you for submitting your work to PLOS ONE and supporting our mission of Open Science.

Kind regards,

Adrian Cyrus Luczon

PLOS ONE

---

## [Decision Letter · Decision Letter 1]

25 Sep 2025

Factors for arboviral seropositivity in children in Teso South Sub County, Kenya

PONE-D-25-31972R1

Dear Dr. Inziani,

We’re pleased to inform you that your manuscript has been judged scientifically suitable for publication and will be formally accepted for publication once it meets all outstanding technical requirements.

Kind regards,

José Ramos-Castañeda, M.Sc., Ph.D

Academic Editor

PLOS ONE

Additional Editor Comments (optional):

Reviewers' comments:

Reviewer's Responses to Questions

**Comments to the Author**

1. If the authors have adequately addressed your comments raised in a previous round of review and you feel that this manuscript is now acceptable for publication, you may indicate that here to bypass the “Comments to the Author” section, enter your conflict of interest statement in the “Confidential to Editor” section, and submit your "Accept" recommendation.

Reviewer #1: All comments have been addressed

2. Is the manuscript technically sound, and do the data support the conclusions?

Reviewer #1: Yes

3. Has the statistical analysis been performed appropriately and rigorously? 

Reviewer #1: Yes

4. Have the authors made all data underlying the findings in their manuscript fully available?

Reviewer #1: Yes

5. Is the manuscript presented in an intelligible fashion and written in standard English?

Reviewer #1: Yes

6. Review Comments to the Author

Reviewer #1: The authors responded appropriately to each of the observations raised, and the corresponding changes were incorporated into the original manuscript. Therefore, I consider the manuscript suitable for publication.

7. PLOS authors have the option to publish the peer review history of their article (what does this mean? ). If published, this will include your full peer review and any attached files.

**Do you want your identity to be public for this peer review?** For information about this choice, including consent withdrawal, please see our Privacy Policy .

Reviewer #1: **Yes: ** Irma Yvonne Amaya-Larios

---

## [Editor Report · Acceptance letter]

PONE-D-25-31972R1

PLOS ONE

Dear Dr. Inziani,

I'm pleased to inform you that your manuscript has been deemed suitable for publication in PLOS ONE. Congratulations! Your manuscript is now being handed over to our production team.

Kind regards,

on behalf of

Dr. José Ramos-Castañeda

Academic Editor

PLOS ONE